# Evaluation of *Monascus purpureus* fermentation in dairy sludge-based medium for enhanced production of vibrant red pigment with minimal citrinin content

**Samira Moradi, Seyed Ali Mortazavi** *

Faculty of Agriculture, Department of Food Science and Engineering, Ferdowsi University of Mashhad, Mashhad, Iran

* Morteza@um.ac.ir

**Data Availability Statement:** All data produced or examined in the course of this study have been incorporated within this published article and adhere to established research protocols.

## Abstract

This study aimed to explore the production of red pigment from *Monascus purpureus* in waste culture medium and its potential health benefits. Subsequently, the *M. purpureus* cultivated in a medium containing dairy sludge as waste, the extracted pigment was purified, and subjected to various analyses, including liquid chromatography mass spectrometry (LCMS) and nuclear magnetic resonance (NMR) to verify its purity, high-pressure liquid chromatography (HPLC) to measure the citrinin levels, microbial, and antioxidant activity. Finally, fermentation was conducted in a batch system using a fermenter. *M. purpureus* was grown in a medium composed of dairy sludge, monosodium glutamate, and glucose, resulting in a biomass yield of 26.15 g/L. After extraction and purification, the sample yielded 4.85 g of dry color. Analysis confirmed the purity of the pigment by LCMS and NMR and revealed low citrinin levels by HPLC. In the fermenter, the sample obtained from enriched culture conditions displayed the highest concentration of monascorubramine, maximum specific growth rate of 0.029/1/h, a cell yield ($Y_{x/s}$) of 0.29 g/g, and a production efficiency of 65% for *M. purpureus*. The produced pigment sample showed potential for use in the food industry due to its low citrinin content and high concentration of red pigment.

## 1. Introduction

Color is a crucial characteristic of food in terms of its appearance, and consumers consider it as an indicator of quality. In recent years, there has been an increasing demand for natural colors due to concerns about the potential negative impact of synthetic colors on human and animal health [1–3].

Microorganism colors are derived from certain species of bacteria, molds, and yeast. Notable examples include species from the genera Monascus, *Paecilomyces*, *Serratia*, *Cordyceps*, *Streptomyces*, and *Penicillium* [4]. Monascus is a member of the Eurotiales order and *Monascus* family, that identified over 20 species of Monascus, with *Monascus pilosus*, *Monascus purpureus*, and *Monascus ruber* being the most significant and widely utilized species in the food industry [5–7]. Pigments produced by Monascus species are widely used in China and East

**Funding:** This work is based upon research funded by Iran National Science Foundation (INSF) under project No. 4005146.

**Competing interests:** The authors declare that they have no competing interests.

Asian countries as natural food pigments in various products such as fish, Chinese cheese, ketchup, beverages like red wine, and meat products such as sausages and hamburgers as a substitute for nitrites [8–10].

Monascus molds produce diverse metabolites, including pigments, monacolins, lovastatin, a toxin called citrinin (monascidine), and gamma-aminobutyric acid. *M. purpureus* is particularly important among the different species due to its high efficiency in pigment production and exhibits lower levels of citrinin, a type of mycotoxin, compared to other species of this mold [11–13]. The presence of citrinin poses significant safety challenges in mold-based products, attracting global attention. Some Asian countries have also introduced their own limits, such as 0.05 μg and 0.2 μg citrinin/g food product in South Korea and Japan, respectively, while the American Food and Drug Administration has set it at 20 μg/kg for agricultural products [5,11]. Various solutions have been proposed to reduce citrinin production, including optimizing culture conditions, incubation temperature, time, dissolved oxygen, and the use of additives such as specific amino acids, and ammonium sulfate [12].

Most studies in this field have focused on the production of edible pigments during the fermentation process [14–16]. However, little has been done to address citrinin production reduction during fermentation of waste. Therefore, the main objective of this project is to minimize citrinin production in waste fermentation medium. Another goal is to explore the use of agricultural wastes, such as dairy sludge, for red pigment production, a topic that has not been extensively researched. Dairy industries, being major producers of wastewater, generate dairy sludge, which contains high levels of organic compounds like carbohydrates and proteins. These can serve as carbon and nitrogen sources for microorganisms and reducing the overall production cost is also crucial for the industry [17]. Pigment production using dairy sludge has not been done yet, and dairy sludge has been used in the production of lactic acid [18] and γ-aminobutyric acid [19]. The utilization of such diverse and readily available resources for pigment synthesis from *M. purpureus* highlights the possibility of sustainable and cost-effective production methods. Various substrates have been evaluated for pigment synthesis through *M. purpureus* fermentation include starch, Saba banana peel, residual beer, cheese whey, soybean meals, waste loquat kernels, date waste substrates, and whey [1].

In this study, dairy sludge was used as a cheap and enriched culture medium and potato dextrose broth (PDB) medium was used as a control. After extracting and purifying the two samples obtained from the control sample and the culture medium containing dairy sludge (enriched sample), the production of pigment was investigated through liquid chromatography mass spectrometry (LCMS) and citrinin by high pressure liquid chromatography (HPLC). Also, the antimicrobial and antioxidant activity of the produced pigment was studied. Finally, pigment production was evaluated in the fermenter.

## 2. Materials and methods

### 2.1. Microorganisms

*M. purpureus* PTCC 5303 mold was obtained from Iran's Industrial Microorganisms Collection Center, grown on a potato dextrose agar (PDA) and incubated at 30°C for 7 days. Then, it was kept in the refrigerator until use, and re-cultivation was prepared from it once every two weeks [20].

### 2.2. Culture media and chemicals

The PDB, PDA and Mueller Hinton agar (MHA) purchased from Sigma Aldrich Co. (Canada). Also, dairy sludge was sourced from dairy factory. All reagents of analytical grade used in the study were purchased from Merck Co. (Germany).

## 2.3. Submerged fermentation

The fermentation process of *M. purpureus* includes several consecutive steps as follows: The spores produced from a 7-day culture of *M. purpureus* are washed from the surface of the culture medium using sterile phosphate buffered salt (PBS). A suspension is prepared with a concentration of $1.5 \times 10^6$ spore/mL (McFarland's 0.5 standard). 1mL of the spore suspension are transferred to a 250 ml flask containing 100 ml of the original PDB culture medium, enriched with dairy sludge (10%v/v), monosodium glutamate (1%w/v), and glucose (10%w/v). The flask is then kept in a dark environment for 14 days in a shaker incubator at a temperature of 30°C and a rotation speed of 160 rpm. The initial pH of the culture medium is adjusted to 5, 6.5, and 8 to stimulate the microorganism to produce pigment [4].

## 2.4. Biomass determination

The fungal biomass was estimated by determining the amount of N-acetylglucosamine released by the acid hydrolysis of chitin, which is present in the mycelium cell wall. Chitin hydrolysis was performed using 10 M HCl in an autoclave at 130°C for 2 h. The hydrolyzed mixture was neutralized to pH = 7, then mixed with acetylacetone reagent, followed by Ehrlich's reagent. Finally, the light absorption of the sample at 530 nm (compared to pure N-acetylglucosamine) was measured [21].

## 2.5. Measurement of extracellular and intracellular pigments

To separate the mycelium, all the contents of the culture medium were filtered using Whatman filter paper. Then, the strained culture medium was centrifuged for 15 min at $7511 \times g$ and prepared solution was diluted with distilled water. The absorption of samples was measured using a visible ultraviolet spectrophotometer at 510 nm (red pigments), 470 nm (orange pigments) and 400 nm (yellow pigments). The amount of produced pigment was obtained by multiplying the absorbance value of the sample by the applied dilution.

The remaining mycelium on the filter paper from the previous step was washed twice with distilled water, then cut into small pieces and added to 10 ml of 70% ethanol (v/v) (pH 2) and kept in a shaker at speed of 120 rpm for 2 h. The ethanolic solution containing intracellular pigment was centrifuged for 15 min at $7511 \times g$ until the mycelium settled and the amount of pigment in the supernatant solution was measured with the mentioned method for intracellular pigment [21].

## 2.6. Isolation and purification of pigment

Fermented broth medium without cells was used for pigment purification. The filtered solution was concentrated using a rotary evaporator (Rotavapor R-210, Buchi, Switzerland) and then lyophilized and powdered. The powder was extracted using hexane (500 ml in total) for 1 h in a shaker (120 rpm) and concentrated using a rotary evaporator under vacuum. The extracted crude pigment was loaded into a silica gel column (60–120 mesh) and followed different ratios of hexane and ethyl acetate were used as detergents. The fractions eluted from the column that were read by spectrophotometer between 300 and 700 nm were combined. Finally, ethanol was used to wash the target compound [22].

## 2.7. Analysis of pigment composition

**2.7.1. Nuclear magnetic resonance (NMR).** The characteristics of the purified pigment were performed by NMR spectroscopy at room temperature using Bruker WM 500 spectrometer [500 MHz ($^1$H NMR)]. A small amount of dried purified pigment was dissolved in 500 ml

of dimethyl sulfoxide (DMSO) and the solution was homogenized. Then, this mixture was analyzed by NMR to evaluate the number of protons ($^1$H) at 500.13 MHz to 125.77 MHz [23].

**2.7.2. LC-MS.**   LC-MS (electrospray ionization, ESI) analysis was done using an Agilent HPLC-MSD series 1100. The sample was first filtered through a 0.2 μm PTFE filter membrane and then placed on the automatic sampler. Next, the pigments were eluted on a Mightysil RP18 column (150 mm × 2 mm Kanto Chemical, Tokyo) using a linear gradient. The gradient ranged from acetonitrile-water containing 0.1% formic acid (60:40, v/v) to acetonitrile-water containing 0.1% formic acid (100:0, v/v). The flow rate was maintained at 0.2 mL/min, the oven temperature was set to 40°C, and running time for the analysis was 25 min. The pigment compounds were detected using electrospray ionization in positive ion mode MS/MS [24].

## 2.8. Citrinin assay

The measurement of citrinin was carried out using HPLC (Milford, USA). Specifically, 20 μl of the purified pigment sample was injected into an HPLC device equipped with a C18 reverse phase column (4.6 mm × 250 mm, 5 μm particle size). The mobile phase used was a mixture of acetonitrile and water (65:35 v/v), and the flow rate was set at 1 mL/min. To detect and quantify citrinin, a fluorescence detector was employed with an excitation wavelength of 331 nm and an emission wavelength of 500 nm [12]. In this method, commercial pure citrinin was used as a standard material and the amount of citrinin was calculated by comparing the absorption rate of the test samples with the standard material sample.

## 2.9. Measuring the antimicrobial activity of pigment

**2.9.1. Well diffusion agar (WDA).**   From the suspension of pathogenic bacteria *Escherichia coli* ATCC 25922, *Pseudomonas aeruginosa* PTCC 1707, *Salmonella typhimurium* PTCC 1609, and *Staphylococcus aureus* ATCC 25923 (obtained from the Center for Biological and Genetic Resources of Iran) with a concentration of half of McFarland, the amount of 10 μL was cultured on MHA medium and then wells with a diameter of 6–8 mm were created in the plates by the end of a sterile pipette and 100 μL of purified red pigment was added to the wells. After placing in an incubator for 48 h at a temperature of 37°C, the diameter of the growth halo around each well was measured [25].

**2.9.2. Minimum inhibitory concentration (MIC) and Minimum bactericidal concentration (MBC) determination.**   In this method, the pigment along with the pathogenic bacteria is placed in a 96-well microplate and after incubation (24 h, 37°C), The first well without turbidity indicating no visible bacterial growth was reported as MIC.

In order to determine MBC, 10 μL were removed from the wells of the 96 microplates in which no color change was observed under sterile conditions and cultured on MHA culture medium. The plates were incubated in the temperature of 37°C, and after 24 h and were examined for growth. The first plate of concentrations cultured from the cell extract in which no colony was observed was considered as the MBC [26].

## 2.10. Investigation of the pigment antioxidant properties

**2.10.1. Inhibition of DPPH radical.**   The sample mixture (the concentration of 1 mg/mL) and DPPH radical solution of 0.2 mmol in 95% ethanol were combined and mixed. After keeping the sample for 30 min at room temperature and in a dark place, its absorbance was recorded at 517 nm and the radical scavenging activity of sample was calculated with following

Eq 1 [27].

$$\text{Radical scavenging activity \%} = (1 - \text{Absorption of sample}/\text{Aabsorption of control}) \times 100 \qquad \text{Eq1}$$

**2.10.2. Ferric ion reducing antioxidant power measurement of the samples (FRAP).** 1 mL of the sample was mixed with 1 mL of distilled water and 1 mL of potassium ferricyanide (1%). After heating at 50°C for 20 min, 2.5 ml of 10% trichloroacetic acid was added to the solution. The resulting mixture was then centrifuged at 750 rpm for 5 min. Next, 2 mL of the supernatant was taken and mixed with 2 ml of distilled water and 1 ml of iron chloride (0.1%). The mixture was stirred and allowed to stand for 10 min at room temperature. Finally, the absorbance of the solution was measured at a wavelength of 700 nm. The percentage reducing activity of samples was calculated using the following Eq 2 [28].

$$\text{Reducing activity \%} = [(1 - \text{Sample absorbance})/\text{Control absorbance}] \times 100 \qquad \text{Eq2}$$

## 2.11. Cultivation in fermenter

One mL of suspension from a 6-day-old PDA slope of *M. purpureus*, grown at 30°C, provided a 1% (v/v) inoculum for shake-flask cultures. For the fermentations, 100 ml of actively growing shake-flask culture grown on identical medium was used. The growth medium was composed of the enriched culture containing the original PDB culture medium with dairy sludge (10%v/v), monosodium glutamate (1%w/v), and glucose (10%w/v). Sterilization of the media was performed at 121°C for 15 min. All medium components were analytical grade and used with distilled water. Shake-flask culture was performed in 250 ml Erlenmeyer flasks containing 100 ml medium incubated at 30°C in an orbital water bath (150 rpm) in darkness. Batch fermenter cultures were performed in a 2–1 glass vessel (Newbrunswick) sealed with a stainless-steel head plate. The working volume was 3L. Agitation was provided at 500 rpm by a flat-bladed impeller and sterile air was supplied at 0.003 L/min. Automatic temperature (30°C) and pH control was performed, the latter using a pH control equipped with a steam-sterilizable pH electrode and sterile solutions of 1 M NaOH or 1 M HCL. A foam controller added silicon antifoam as required. The glass vessel was autoclaved at 121°C for 15 min and sterile medium was added aseptically upon cooling. After inoculation (1%, v/v), the fermenter was maintained in darkness. After 36 h, feeding with a carbon source (glucose) was initiated. The feeding lasted for 8 h at a rate of 0.03 L/min. The fermentation lasted for a total of 134 h.

## 2.12. Statistical analysis

Data were analyzed by Minitab software (version 20) via one-way ANOVA and Tukey test at confidence level of 95% ($p < 0.05$) was applied to determine differences between data means, so it was necessary to note that experiments were conducted at three replications.

## 3. Result and discussion

### 3.1. The results of biomass production in the culture environment in various pH

The treatment was investigated in three pH ranges of 5 (22.5 ± 0.3 g/L), 6.5 (26.75 ± 0.5 g/L) and 8 (24.3 ± 0.7 g/L). According to the obtained results, the maximum amount of biomass was obtained at pH 6.5 (26.75 g/L) and they have a significant difference with each other (p-value < 0.05). So, initial pH was fixed at 6.5.

The pH of the fermentation environment is an important factor in the synthesis of red pigment by Monascus due to high pH values lead to the chemical change of orange pigments to extracellular and water-soluble red pigments [29]. Usually, at low pH, more yellow pigment is produced, and at higher pH, red pigment gradually dominates [5,30]. The pH effect on pigment produced by *M. purpureus* in submerged fermentation was evaluated and pH can affect the stability of the pigment, which is lost at lower pH values. It was also found that the red pigment compared to the yellow pigment and orange is more sensitive to pH [30]. In different studies, it has been reported that *M. purpureus* has more red pigments at pH values of 6–8 [22], and range of 5.5–8.5 [11]. According to the report, red pigment production is positively affected by high pH values and high concentrations of monosodium glutamate. This means that increasing the pH level and adding higher amounts of monosodium glutamate can lead to an increase in the production of red pigments [22]. On the other hand, the transfer of water-soluble extracellular pigments from the cell to the fermentation medium is hindered when the pH values are low. This implies that at lower pH levels, the ability of the pigments to move out of the cells and into the surrounding medium is limited [31].

## 3.2. Pigment extraction and purification

The extraction and purification of two pigment samples obtained from the control sample and the culture medium containing dairy sludge (enriched sample) (Fig 1) was done. According to the pictures of the samples, it can be seen that there is more color in the sample obtained from

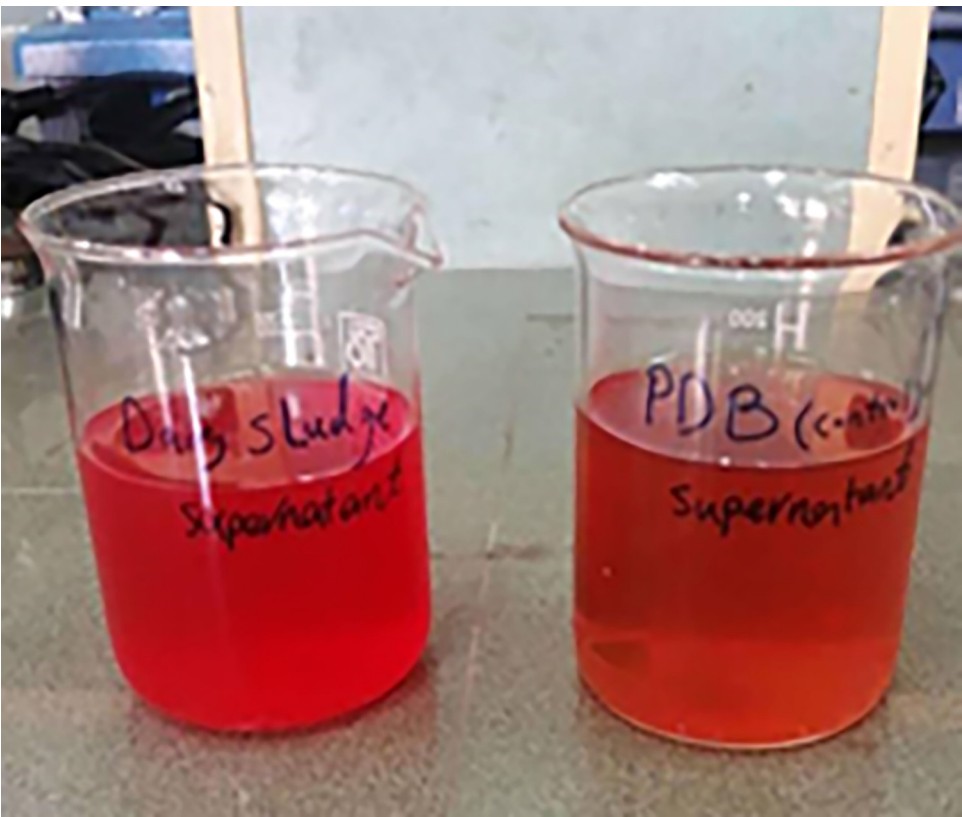

**Fig 1.** Image related to dry pigment produced by the strain in enriched medium (left) and control (right) samples.

the enriched culture medium containing dairy sludge. The amount of 37 and 19 absorbance units (AU500)/mL for the red pigment of the enriched and control samples was obtained, respectively. Also, the amount of 4.85 and 2.5 g of dry color/L was obtained for the enriched and control sample, respectively, after extraction.

The LC-MS spectrum of the samples is depicted in Fig 1A and 1B. The peaks corresponding to the pigment composition in enriched and control samples are observed within the retention time range of 5.5–8 and 4.5–8 min, respectively (Fig 2). The purity of the pigment was determined to be 91.9% and 85.5% for enriched and control samples, respectively. The peaks in the enriched and control samples are very similar to each other, and the small shift in the retention time is due to the presence of more impurities in the control sample.

The type of medium used in the fermentation process has been consistently found to be a crucial factor in pigment production and metabolic products, including pigments, are known to be dependent on the culture conditions [32]. In this study, the use of dairy sludge as waste, which has large amounts of nitrogen, carbon and minerals, had a positive effect on pigment production and had a synergistic effect with the main culture medium of *M. purpureus*. Other studies have also demonstrated the influence of different culture media on pigment production. For instance, the culture medium based on whey resulted in a red pigment production efficiency of 1.12 UA510 with *M. purpureus* [29]. Similarly, a pigment amount of 22.25 UA500 was reported in a culture medium containing monosodium glutamate, nitrogen, and waste [4]. In the studies conducted for the production of pigment by *M. purpureus*, the compounds of sucrose esters [9] and corn starch with oils [33] evaluated in fermentation medium. Similarity, high extracellular pigment production of 34.12 U/ml was observed in submerged fermentation using a glucose-based medium with *M. purpureus* [11].

On the other hands, Studies have highlighted the significance of specific compounds, such as monosodium glutamate, in improving pigment production, particularly red pigments [34]. For example, it was found that jackfruit seed powder alone does not produce water-soluble pigments, but the addition of sodium magnesium glutamate enables the production of red water-soluble pigment [2].

## 3.3. Evaluation of pigment structure by $^1$H NMR

In the Fig 3, a distinct peak at 1.2 ppm was observed solely in the control sample, indicating a potential impurity in the enriched sample (Fig 3A). Both spectra exhibited a prominent peak around 1 ppm, corresponding to $RCH_3$ (alkyl) groups (Fig 3). The subsequent peak in the 2–3 ppm range likely originated from $RC = OCH_3$ functional groups. Multiple peaks in the 3–4 ppm range were observed, possibly attributable to $RCH_2OH$, $RCH_2OR$, and $RC = OOCH_3$ groups. Furthermore, two doublets appeared at 4.3 ppm and 4.9 ppm, indicating the presence of RNH and protons of amine groups as well as alcohol [35]. Data concludes that it was unsaturated aliphatic nitrogenous compound containing ɣ-lactone ring. The molecular formula of the compound is $C_{21}H_{29}NO_4$. The structure of the pigment has numerous similarities with two classical red pigments, rubropunctamine and monascorubramine [22]. *Monascus* spp. are responsible for the production of the primary color pigments, which exhibit the six major pigments consist of rubropunctamine ($C_{21}H_{26}NO_4$) and monascorubramine ($C_{23}H_{27}NO_4$) in red, rubropunctatin ($C_{21}H_{22}O_5$) and monascorubrin ($C_{23}H_{26}O_5$) in orange, and monascin ($C_{21}H_{26}O_5$) and enkaflavin ($C23H30O5$) in yellow [29]. Also, in a study, $^1$H NMR spectral patterns at 0.87, 1.26, 1.28, 1.63 and 2.03 ppm readily indicated the presence of long alkyl chain attached to a carbonyl and a broad peak was observed at 9.96 ppm that implied presence of an aldehyde and two carbonyl groups (Mondal, Pandit, Puttananjaiah, Harohally, & Dhale, 2019).

a

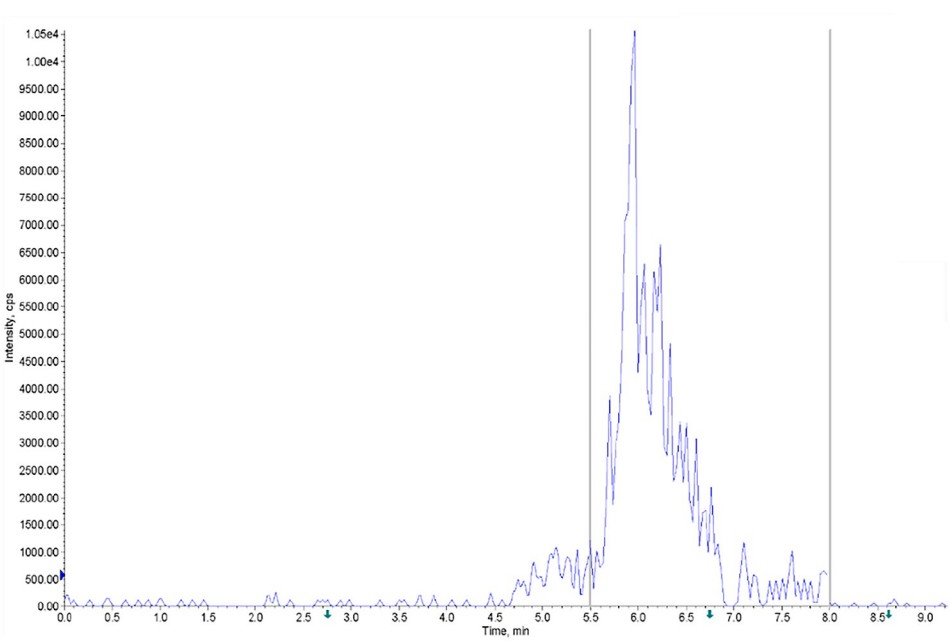

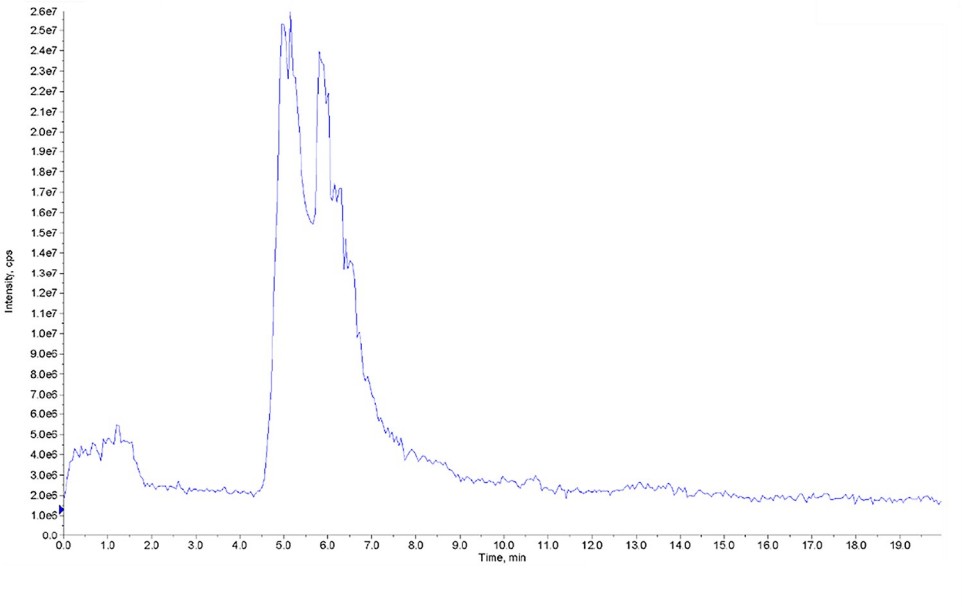

b

**Fig 2.** LCMS spectra of obtained pigment samples from (a) enriched and (b) control fermentation medium.

a

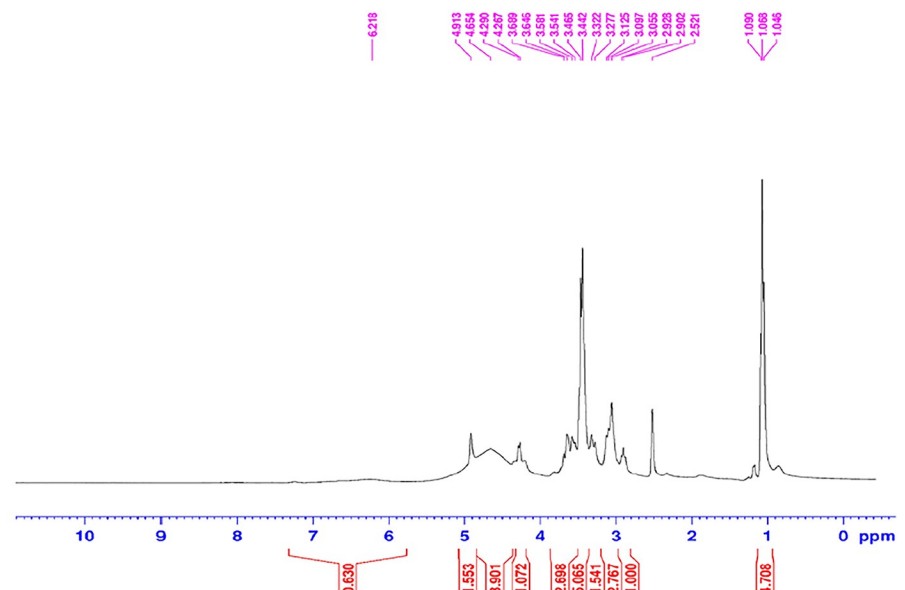

b

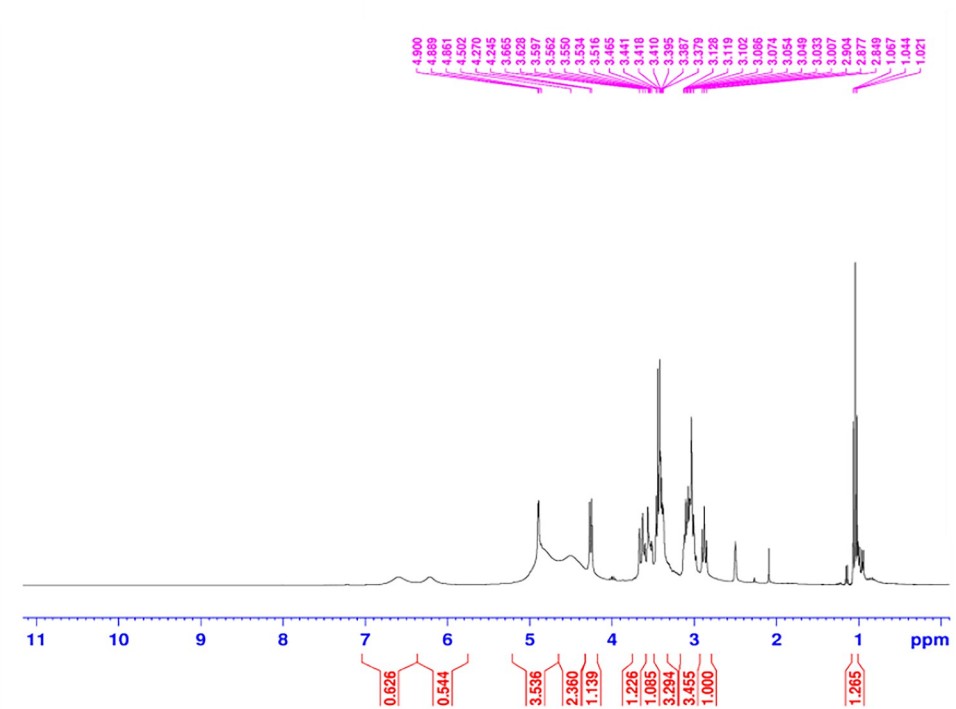

**Fig 3.** [1]HNMR spectra of obtained pigment samples from (a) enriched and (b) control fermentation medium.

### 3.4. Citrinin measurement by HPLC

The obtained results indicate a significant reduction in the amount of citrinin in the enriched sample compared to the control sample. The enriched sample had a citrinin content of less than the standard limit of 0.05 ppm, while the control sample had a citrinin content of 2.5 ppm (Fig 4). In general, an 80% reduction in citrinin has been observed in this environment resulting from the waste compared to the control environment, which compared to similar studies, a significant reduction in the amount of citrinin has been obtained. In a study analyzing citrinin and pigment production in a culture medium containing tyrosol, it was observed that the addition of tyrosol led to a decrease in citrinin content by approximately 51.5% compared to the control medium without tyrosol [6]. The use of millet in the fermentation culture was found to decrease citrinin production in the pigment produced by *M. purpureus* [36]. Similar results were observed in studies involving flavonoids such as apigenin, genistein, rutin, alpha-glucosylrutin, or troxerutin, which strongly reduced citrinin synthesis while increasing pigmentation [37,38]. Zhen et al. (2019) reported that NaCl inhibits citrinin synthesis but stimulates the synthesis of pigments [39]. It is worth noting the biosynthesis that pathways of pigments and citrinin are related, but the results of the experiments suggest that these pathways are independent among different Monascus species under the conditions of this study [40]. Overall, these findings demonstrate the potential to manipulate culture conditions and add specific compounds to regulate citrinin production and enhance pigment production in Monascus species.

### 3.5. Comparative tests of pigment antimicrobial effect

The results of the WDA test, as shown in Table 1, indicate that both the control and enriched samples exhibited the highest resistance (lowest halo size) against *E. coli*, while they showed the highest sensitivity (largest halo size) against the *S. aureus* strain. The results of the MIC and MBC tests, presented in Table 1, reveal that *E. coli* was the most resistant strain, while *S. aureus* was the most sensitive strain to both samples. These findings align with the results of the WDA test.

Overall, the dairy sludge culture medium exhibited a significantly stronger antimicrobial effect against pathogenic microorganisms compared to the control sample. Pigments have long been used as natural and safe colors in Asian countries due to their excellent coloring and antibacterial properties [8]. Reported that extracts of fungal colorants have also been shown to possess antibacterial activities against pathogenic bacteria such as *S. aureus* [41,42]. According to a study, the antibacterial activity of the Monascus pigment was found to be higher than that of the commercial red pigment against all tested bacteria. The MBC for *Bacillus cereus* ATCC11778 was determined to be 256 mg/mL for the commercial pigment, while it was lower at 128 mg/mL for the Monascus pigment [26].

It is worth noting that Gram-negative bacteria, including *E. coli*, have a complex cell membrane structure consisting of lipopolysaccharides and two layers of phospholipids. This outer layer acts as a barrier, making it more challenging for compounds to penetrate the cell membrane and exert their antimicrobial effects, which may explain the higher resistance observed in *E. coli* compared to *S. aureus* [18].

### 3.6. Measurement of antioxidant activity through inhibition of DPPH radicals and FRAP test

The enriched sample exhibited 55.26% and 0.105 μmol/L of antioxidant activity to inhibit DPPH radicals and FRAP reducing power, respectively. In comparison, the control sample showed 32.4% and 0.061 μmol/L for DPPH radicals and FRAP reducing power, respectively.

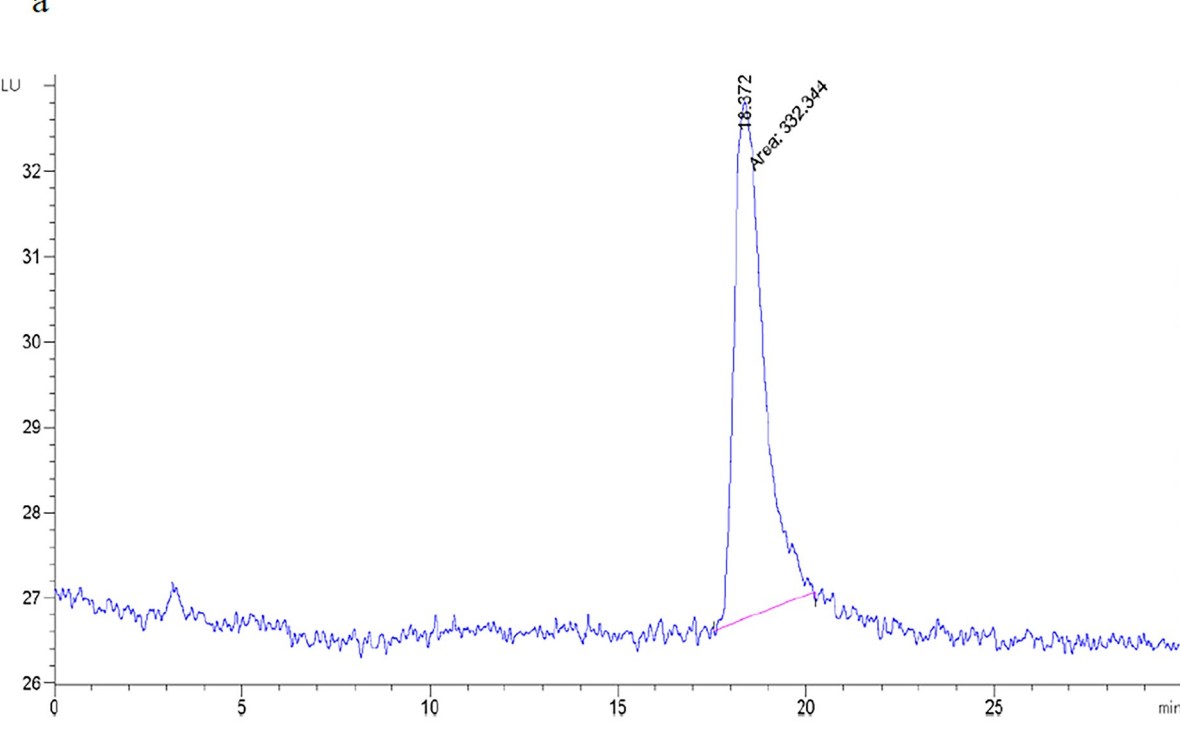

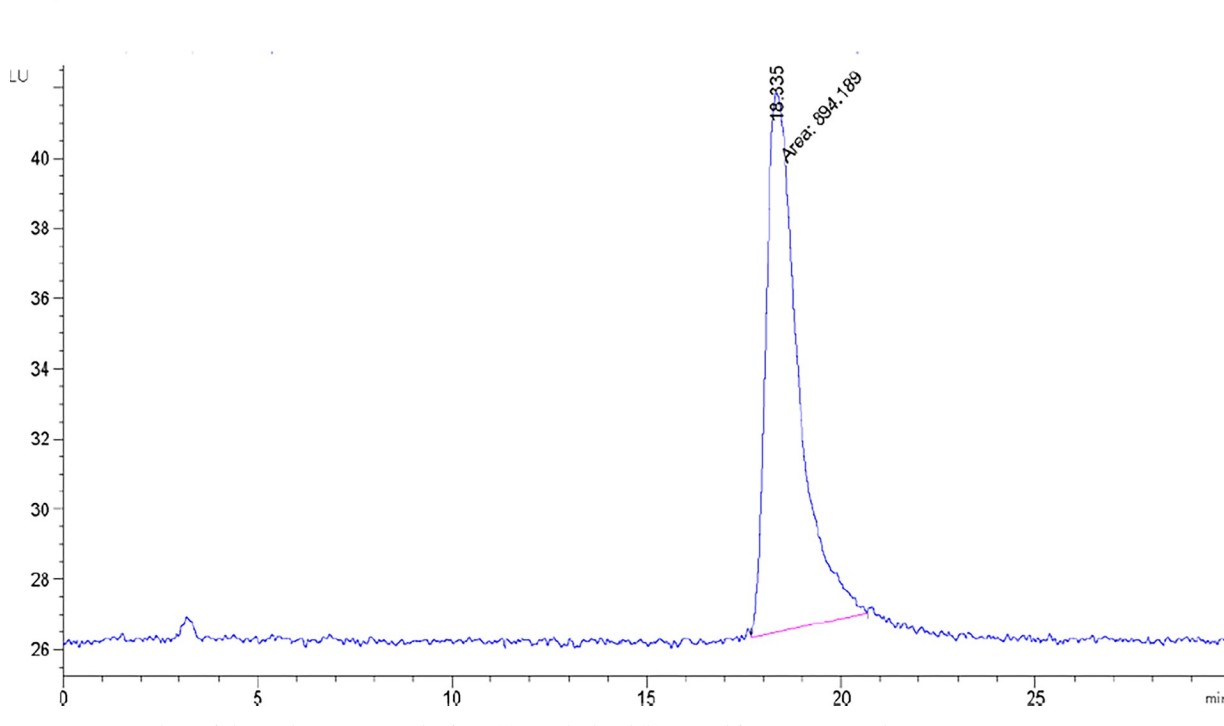

**Fig 4.** HPLC analysis of obtained pigment samples from (a) enriched and (b) control fermentation medium.

**Table 1. Results related to the evaluation of pigment antimicrobial activity (WDA, MIC and MBC) of obtained samples from control and enriched culture media.**

| Strain | WDA (mm) | | MIC (mg/ml) | | MBC (mg/ml) | |
|---|---|---|---|---|---|---|
| | Optimal sample | Control sample | Optimal sample | Control sample | Optimal sample | Control sample |
| *S. typhi* | 12.00 ± 0.48[cA] | 8.00 ± 0.23[dB] | 125±5[bB] | 500±10[aA] | 125±6[bB] | 500±12[aA] |
| *E. coli* | 10.00 ± 0.50[dA] | 9.00 ± 0.21[cB] | 250±7[aB] | 550±10[bA] | 250±5[aB] | 550±5[bA] |
| *L. innocua* | 14.00 ± 0.46[bA] | 10.00 ± 0.32[bB] | 125±5[bB] | 500±10[aA] | 125±7[bB] | 500±5[aA] |
| *S. aureus* | 19.00 ± 0.45[aA] | 11.00 ± 0.24[aB] | 100±5[cB] | 250±8[cA] | 100±5[cB] | 250±10[cA] |

Small letters a to b indicate significant differences between different strains in each column and capital letters A to B indicate significant differences between different samples in each row.

The pigment sample obtained from the dairy sludge culture medium displayed approximately twice the antioxidant activity compared to the control sample, which can be attributed to the presence of more red pigment in the enriched culture medium. Similarity, the pigment demonstrated antioxidant activity against the 2,2′-azino-bis (3-ethylbenzothiazoline-6-sulfonic acid) radical, with an IC50 of 14.42 μg/mL [41]. Another study involving the pigment monascin, produced by *M. purpureus*, reported DPPH (26%), H2O2 (77%), and reducing power (0.57 AU) activities at concentrations of 37.5, 40, and 46.15 μg/mL, respectively [23]. Pigments produced by *Monascus* sp. belong to the azaphilone family, which are cyclic compounds with at least one chiral center. These azaphilones exhibit a wide range of biological activities, including anti-obesity effects such as adipogenesis and lipolysis, as well as anti-cancer, anti-inflammatory, anti-depressant, anti-osteoporosis, and anti-diabetic effects [8,43,44]. The components of the pigments, reduces endothelial adhesion induced by reactive oxygen species (ROS) formation, TNFα and NF-κB activation. It also reduces amyloid-β toxicity and oxidative stress in caenorhabditis elegans by increasing the activities of SHSP-16, glutathione S-transferase, and superoxide dismutase [2].

### 3.7. Biomass and pigment production kinetics in fermenter

Fig 5A illustrates the increase in biomass until 124 h of fermentation that indicates successful growth and utilization of nutrients. The highest amount of biomass was produced in the

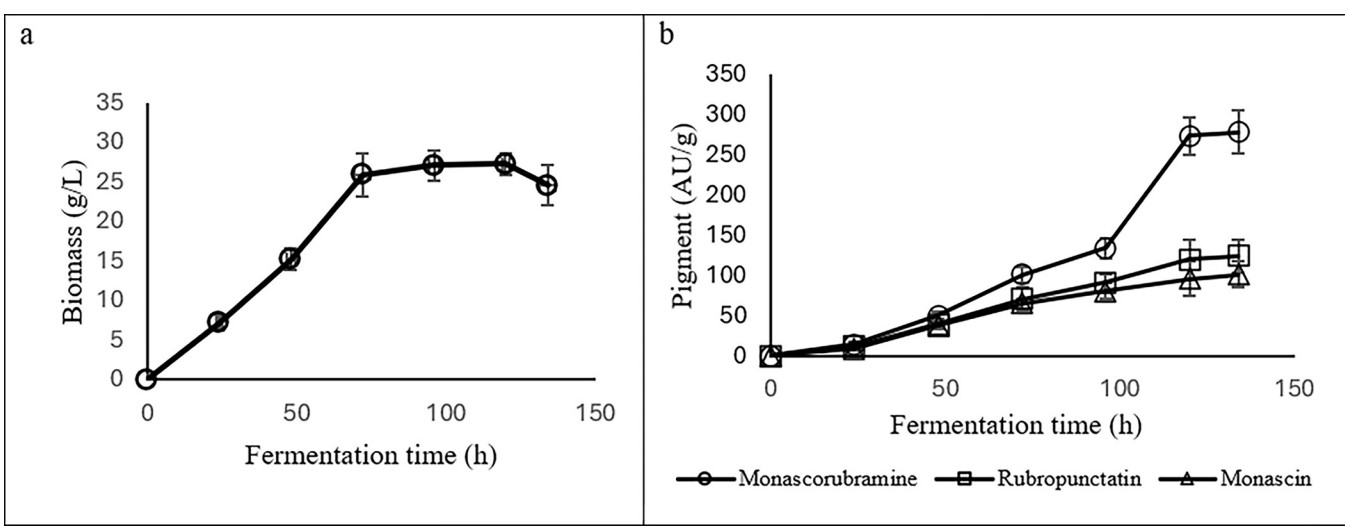

**Fig 5.** Kinetics of (a) biomass production and substrate consumption and (b) pigment production by *M. purpureus* in batch fermenter during 134 h in enriched culture environment.

stationary phase (72–120 h). However, after 124 h, biomass production declined. This could be attributed to the accumulation of toxic metabolites in the environment, leading to the death phase. Fig 5B shows the kinetics of pigment production that in the first 48 h, there was no significant difference in the amount of the three pigment compounds. After 48 h, the red monascorubramine pigment showed a significant (p-value < 0.05) increase compared to the other two pigments. The highest amount of color was also produced in the stationary phase (96–120 h).

To enhance pigment production in Monascus species, optimizing cultivation conditions is crucial. Factors such as inoculum size, temperature, initial pH, oxygen concentration, and nutritional components (nitrogen, carbon, and minerals) can be adjusted. Previous studies have shown similar results in optimizing cultivation conditions [45,46]. Glucose has been found to be a favorable carbon source for higher biomass production compared to other sources. Additionally, glucose concentration influences the production of yellow and red pigments. Higher glucose concentrations tend to shift the maximum absorption towards red pigments. Also, the carbon-to-nitrogen ratio also plays a role in stimulating pigment production in filamentous fungi [5,22]. Specific studies have determined optimal conditions for red pigment production by *M. purpureus*. These conditions include a 2% inoculation rate (v/v), carbon source of 75 g/L lactose, nitrogen source of 25 g/L monosodium glutamate, and a pH of 7. Under these conditions, the maximum red pigment production was reported as 38.4 AU at 510 nm [29]. The utilization of dairy sludge as a culture medium has been shown to significantly impact pigment production. In line with the results of this study, optimizing γ-Aminobutyric acid production in dairy sludge medium, the γ-Aminobutyric acid production reached 359.45 ppm [19].

### 3.8. Determination of kinetic parameters in fermenter

The maximum specific growth rate ($\mu_{max}$) for *M. purpureus* was determined to be 0.029 1/h. The efficiency of cell production compared to the substrate ($Y_{x/s}$) was found to be 0.29 g/g, and the specific rate of biomass production ($q_s$) was calculated as 3.1 g/g.day (Table 2).

In a similar study, the specific growth rate of *M. purpureus* on whey was determined to be 0.023/1/h, with a maximum pigment production efficiency of 4.55 AU [29]. In line with results of this study, the productivity values of 0.059, 0.072, and 0.032 AU/h, respectively, and the specific growth rate values as 0.03 0.04, and 0.017 1/h for lactose, glucose, and hydrolyzed lactose were reported, respectively [47]. The red pigment produced with *M. purpureus* after 192 h of fermentation with productivity and specific growth rate as 2.3 UA/h and 0.03/1/h, respectively [48]. In another study, the addition of glucose to a rice paddy-based culture medium resulted in a efficiency of 21.2 U/mL in 336 h ($q_s$) and specific growth rate 0.06 U/mL/h [11]. These findings indicate that the chosen culture medium in this study is highly productive and provides a stable substrate for the production of red pigment by *M. purpureus*.

## 4. Conclusion

Wastes, especially dairy sludge, are of particular importance in the production of metabolites due to the fact that they contain nutrients necessary for the growth of microorganisms and

**Table 2. Kinetics parameters of *M. purpureus* fermentation in batch fermenter during 134 h in enriched culture environment.**

| Kinetic parameters | Responses |
| --- | --- |
| $Y_{x/s}$ (g/g) | 0.29±0.01 |
| $q_s$ (g/g/day) | 3.1±0.22 |
| $\mu_{max}$ (l/h) | 0.029±0.003 |

their low price, which had a significant effect on the production of pigments in this study. These pigments have diverse applications in the food industry, including coloring, preserving, flavoring, and functional food additives. In this study, the cultivation of *M. purpureus* in a medium containing dairy sludge, monosodium glutamate, glucose, and PDA resulted in increased biomass and pigment production compared to the control medium. The study emphasizes the importance of selecting the appropriate culture medium, cultivation methods, extraction, and purification techniques to ensure the production of pigments with suitable purity, minimal citrinin content, and practical characteristics. Fermentation in the fermenter leading to higher red pigment production and favorable kinetic characteristics. Based on these findings, the method described in the study and the used wastes have a high potential for industrial production and the use of manufactured pigments in different food products. The resulting color shows a promising prospect for its application in the food industry.

## Supporting information

**S1 File.**
(ZIP)

## Author Contributions

**Conceptualization:** Samira Moradi.

**Data curation:** Samira Moradi.

**Formal analysis:** Samira Moradi.

**Investigation:** Samira Moradi, Seyed Ali Mortazavi.

**Resources:** Samira Moradi.

**Writing – original draft:** Samira Moradi.

**Writing – review & editing:** Samira Moradi.

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
