## [Decision Letter · Decision Letter 0]

29 Oct 2024

PONE-D-24-30978Optimizing Monascus purpureus Fermentation in Dairy Sludge-Based Medium for Enhanced Production of Vibrant Red Pigment with Minimal Citrinin ContentPLOS ONE

Dear Dr. Mortazavi,

Thank you for submitting your manuscript to PLOS ONE. After careful consideration, we feel that it has merit but does not fully meet PLOS ONE’s publication criteria as it currently stands. Therefore, we invite you to submit a revised version of the manuscript that addresses the points raised during the review process.

We look forward to receiving your revised manuscript.

Kind regards,

ARNABJYOTI DEVA SARMA

Academic Editor

PLOS ONE

Journal Requirements:

Additional Editor Comments:

Dear Author,

Thank you for submitting your manuscript to PLOS ONE. After the initial review, it is requested to you to kindly provide the graphical abstract of the manuscript along with good image quality of figure no 1 with outmost important.

Anticipating your kind response.

Regards,

Dr. Arnabjyoti Deva Sarma

Academic Editor

PLOS ONE
---

## [Author Response · Author response to Decision Letter 0]

8 Nov 2024

Dear editor 

Thank you for considering our manuscript worthy of peer review in PLOS ONE. 

The points considered by the respected editor were made in the manuscript as follows:

Questions Responses

Please review your reference list to ensure that it is complete and correct. If you have cited papers that have been retracted, please include the rationale for doing so in the manuscript text, or remove these references and replace them with relevant current references. Any changes to the reference list should be mentioned in the rebuttal letter that accompanies your revised manuscript. If you need to cite a retracted article, indicate the article’s retracted status in the References list and also include a citation and full reference for the retraction notice. None of the articles were retracted and the references of the manuscript were not changed.

After the initial review, it is requested to you to kindly provide the graphical abstract of the manuscript along with good image quality of figure no 1 with outmost important. The graphical abstract was drawn and uploaded A higher quality figure was added as Figure 1 in the manuscript.

While revising your submission, please upload your figure files to the Preflight Analysis and Conversion Engine (PACE) digital diagnostic tool, https://pacev2.apexcovantage.com/.

The figures were uploaded on the mentioned website and the figures provided by the software were of lower quality than the original figures, so the initial figures of the manuscript were placed in the file.

Dear reviewer

Thank you for carefully reviewing the manuscript and providing your comments.

Questions Responses

This type of research needs special statistical design. Since statistical optimization was not used in this project, the word optimization was removed from the title and changed to “Evaluation of Monascus purpureus Fermentation in Dairy Sludge-Based Medium for Enhanced Production of Vibrant Red Pigment with Minimal Citrinin Content”.

Why italic?? The manuscript was corrected.

Monascis based product or Mold base prosuct It is more correct to use mold-based products, so the text was changed to mold-based products.

Is this comparison logic as one is solid but the second is liquid??? The manuscript was corrected.

PDA was changed to PDB due to typographical error.

Which sample3??? two samples obtained from the control sample and the culture medium containing dairy sludge (enriched sample).

Is there any PTCC M. purpureus PTCC 5303

Please check (starch yeast powder) The purpose was of potato dextrose agar (PDA) and manuscript was corrected.

Are you sure after 7 day spore produced?? According to the used article, 7 days of incubation was done.

Comparison of Monascus purpureus growth, pigment production and composition on different cereal substrates with solid state fermentation

panelIgnatius Srianta, Elok Zubaidah, Teti Estiasih, Mamoru Yamada, Harijono

How you adjusted this concentration???? a concentration of 1.5 x 10 6 spore/mL (McFarland's 0.5 standard) was used and the manuscript was corrected.

How you transform absorption data to absolute content of pigment?? The amount of produced pigment was obtained by multiplying the absorbance value of the sample by the applied dilution (AU/ml).

There is not any method of purification?? Pigment purification was done as follows, which was explained in section 2-6:

Isolation and purification of pigment

 Fermented broth medium without cells was used for pigment purification. The filtered solution was concentrated using a rotary evaporator (Rotavapor R-210, Buchi, Switzerland) and then lyophilized and powdered. The powder was extracted using hexane (500 ml in total) for 1 h in a shaker (120 rpm) and concentrated using a rotary evaporator under vacuum. The extracted crude pigment was loaded into a silica gel column (60-120 mesh) and followed different ratios of hexane and ethyl acetate were used as detergents. The fractions eluted from the column that were read by spectrophotometer between 300 and 700 nm were combined. Finally, ethanol was used to wash the target compound.

Mukherjee G, Singh SK. Purification and characterization of a new red pigment from Monascus purpureus in submerged fermentation. Process Biochemistry. 2011;46(1):188-92.

How to determine the quantity of Citrinin? In this method, commercial pure citrinin was used as a standard material and the amount of citrinin was calculated by comparing the absorption rate of the test samples with the standard material sample.

---

## [Editor Report · Decision Letter 1]

20 Nov 2024

Evaluation of Monascus purpureus Fermentation in Dairy Sludge-Based Medium for Enhanced Production of Vibrant Red Pigment with Minimal Citrinin Content

PONE-D-24-30978R1

Dear Dr. Mortazavi,

We’re pleased to inform you that your manuscript has been judged scientifically suitable for publication and will be formally accepted for publication once it meets all outstanding technical requirements.

Kind regards,

ARNABJYOTI DEVA SARMA

Academic Editor

PLOS ONE
---

## [Editor Report · Acceptance letter]

29 Nov 2024

PONE-D-24-30978R1 

PLOS ONE

Dear Dr. Mortazavi, 

I'm pleased to inform you that your manuscript has been deemed suitable for publication in PLOS ONE. Congratulations! Your manuscript is now being handed over to our production team.

Kind regards, 

on behalf of

DR. ARNABJYOTI DEVA SARMA 

Academic Editor

PLOS ONE